# A Study on Practicing Qigong and Getting Better Health Benefits in Biophilic Urban Green Spaces

**Shih-Han Hung** [1], **Wan-Yu Chou** [2] and **Chun-Yen Chang** [1,*]

1   Department of Horticulture and Landscape Architecture, National Taiwan University, Taipei 10673, Taiwan; shellyhung6327@gmail.com
2   Graduate Institute of Landscape Architecture and Recreation Management, National Pingtung University of Science & Technology, Pingtung 912301, Taiwan; wychou@mail.npust.edu.tw
*   Correspondence: cycmail@ntu.edu.tw

**Abstract:** In natural spaces, people experience traditional environmental Qi (TEQ), which supports healthy environmental energy flow, and helps them gain an overall improved Qi experience from practicing Qigong. However, what kind of urban green spaces support Qigong? This study provides an analysis that measures TEQ, Qi experience, flow experience, restorative experience, and preference when practicing Qigong in different urban green spaces. A total of 654 valid data points were collected. The results indicate that subjects practicing "breathing" among trees, meadows, and waterscapes perceived higher TEQ, Qi experience, flow experience, and restorative experience, and preferred it to the environment of plazas. In addition, practicing Qigong in environments featuring biophilic elements, such as plants (meadows and trees), elicits flow experience and Qi experience in the built environment. Water, an important biophilic element, also produces better TEQ and restorative experiences, and is preferred by human beings. These results make a connection between Qigong, experiences, and biophilic urban green spaces, and offer suggestions for users to gain health benefits while exercising in urban areas.

**Keywords:** biophilic elements; traditional environmental Qi (TEQ); Qi experience; flow experience; health benefits

## 1. Introduction

Traditional environmental Qi (TEQ) is an invisible substance that is present throughout the environment and is hard to measure. To illustrate this point, scholars have noted that a compatible environment may influence the practice of mind–body exercise [1–4]. One mind–body exercise, Qigong, integrates a harmonious interaction between humanity and the environment. When a human being is in a compatible environment, they may feel a special magnetic field, which allows their body to resonate with the environment [1–4]. The more people are predisposed to a place, the more benefits they feel [5]. Chou, Hung, and Chang [6] interviewed people who are sensitive to Qi, exploring the intensity of quality associated with different natural spaces and the factors that predict the environmental quality of landscape configuration and structures, attempting to measure the factors that influence TEQ. Another study found that on an urban green campus, large continuous grasslands and waterscapes produce better health benefits than areas with dense trees and large artificial structures [7].

In addition, it has been suggested that good TEQ may induce an inner feeling of Qi experience and flow experience. Flow experience is a state of human–environment interaction that plays a significant role in activities, creating an optimal psychological experience [8,9]. Natural places and visual experiences provide physical content and settings for nature-based activities, such as hiking, bushwalking, and rock climbing, which induce flow experiences [8,9]. A compatible environment leads to recovery and reflection experiences, and influences flow experience [10,11]. Williams and Harvey [11] described the features

of "fascination," "compatibility," and "beauty" in a forest environment, which trigger a sense of connection with the environment and lead to flow experiences. The higher the level of relaxation and the more features of "fascination," the deeper the flow experience may be. Moreover, it has also been found that people's preferences for the environment and activities may influence their flow experience [11–13].

Relevant to issues of interaction with the environment, a growing body of research on contact with nature or green space has supported our understanding of the psychological and physiological health benefits of experiences in nature. Much of Western science's current understanding of the human–environment relationship is based on (a) attention restoration theory (ART) and preference [14], and (b) stress reduction theory (SRT) [15], both of which provide a solid support for a link between the natural environment and health [16–18]. The biophilia hypothesis [19] explains why humans initiate a connection to nature; biophilic design [20] is a method of using natural elements, such as water, plants, and natural processes, as design strategies in the built environment to enhance human exposure to nature. These design strategies provide stress release and have physical and psychological health benefits [21,22]. Both stakeholders and citizens are concerned with using nature-based solutions in urban areas in ways that connect biodiversity, ecosystems, blue-green infrastructures, and human physical and psychological health to social and environmental benefits, such as incorporating green infrastructure, biophilic design, urban green space, and so forth [23]. Moreover, in an empirical study, Herzog and Kropscott [24] used a forest setting as visual access to predict a preference matrix for landscape preference. Tang et al. also found that a strong connection to nature could help people feel more restorativeness and that people in their study had a stronger landscape preference for the rural forest landscape [25]. Compared to urban environments devoid of water or plants (i.e., biophilic design elements), those elements found in the natural environment or urban green spaces could attract people's attention, evoke positive emotions, and invoke a state of physical and psychological relaxation [15,16,26]; a field or forest landscape provides better restorativeness than an urban one [27]. Furthermore, people feel more restorativeness and physical stress release in the openness of a meadow and lake landscape in an urban park than in the openness of a paved plaza [28]. This is in line with the savanna hypothesis, which posits that humans prefer openness and that spatial landscapes with scattered trees could provide a positive affiliation [29].

Qigong is a system of movement that fosters consciousness control skills. Through holistic sensory perception, which involves visual experience and feelings, we may evaluate our preferences and restore mental fatigue through nature. The natural arrangement of objects in the environment might induce a flow experience. Although substantial studies have been performed on how the natural environment improves the restorativeness experience and how people generally have a landscape preference for non-urban landscapes, studies exploring Qigong practitioners' psychological states in different environments are still lacking. Therefore, based on available knowledge of the impact of landscapes on human health, we would like to ask what kind of urban green spaces support Qigong practitioners? Are urban natural landscapes better than plaza-based environments? This approach seeks a deeper understanding of the health benefits of the landscape from the perspective of Qigong practitioners and aims to fill this gap in knowledge about TEQ, Qi experience, flow experience, and psychological outcomes, and so forth in urban green spaces.

## 2. Materials and Methods

### 2.1. Area Description

National Taiwan University (NTU) is a green urban environment that comprises various ecological systems, landscape types, evergreen plants, and flowers, including many biophilic elements. It is also a popular place for nearby residents to exercise, explore, and relax. Based on Chou et al.'s [6] interviews of mind–body experts who walked around campus to identify the environmental attributes that affect their Qi experience, we selected choice areas of the campus to be our experimental sites. These included

seven landscape types, categorized into biophilic elements and patterns [20] of those scenes in green urban spaces. These environmental attributes, which include water, plants, natural materials, spaciousness, and so forth, could bring significant benefits to human health. In addition, we observed people practicing Qigong in several places on campus, including near waterscapes, meadows, and plazas with trees. In this study, we focused on seven main types of landscape on the NTU campus—waterscapes, meadows, tree-covered, plazas, plazas with waterscapes, plazas with meadows, and plazas with trees—to research the relationship between psychological experiences and environments. The 30 experimental sites were categorized by the dominant percentage of visual elements in the environment. Figure 1 shows the samples of these seven main landscape types at NTU.

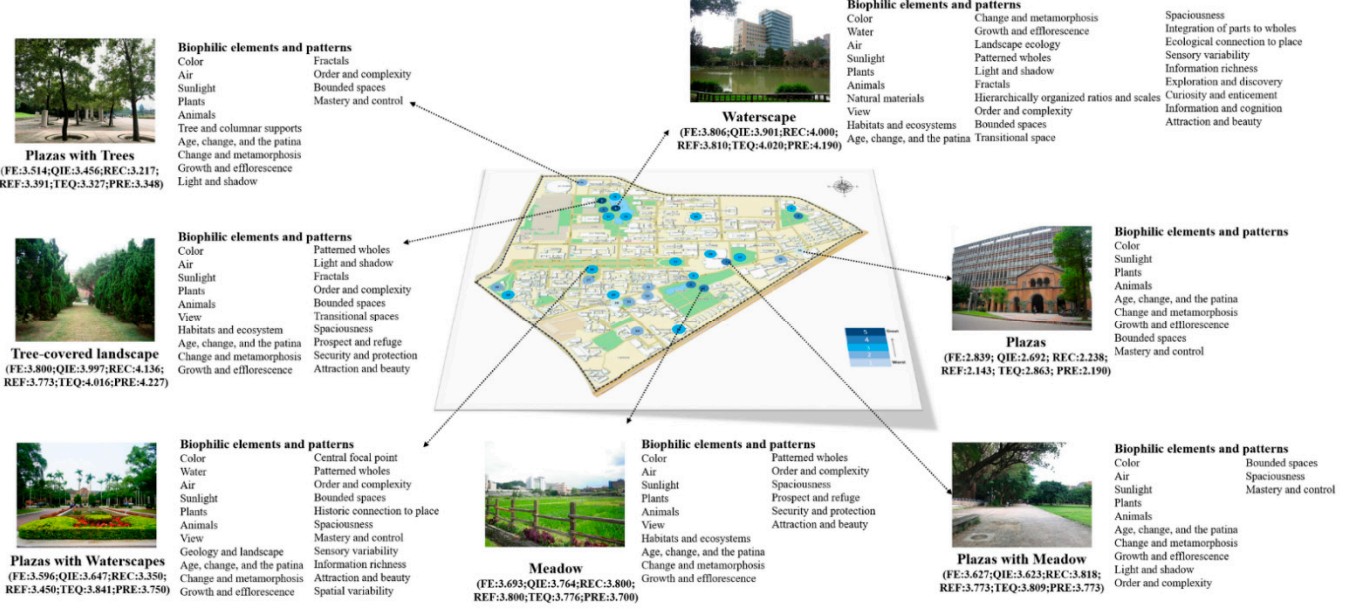

Note: FE: Flow experience; QE: Qi experience; REC: recovery; REF reflection; TEQ: Tradition environmental Qi; PRE: preference; All photos were took by researchers.

**Figure 1.** Map of the experimental sites with biophilic elements and patterns in urban green spaces.

## 2.2. Factors Measured by Questionnaires

### 2.2.1. Traditional Environmental Qi (TEQ)

The concept of TEQ deeply relies on arrangements of landscape, weather, distribution, and the overall feeling of Qi. In other words, TEQ uses the human being as a sensor to feel the flow of Qi in the environment and capture holistic environmental aspects in the interactions between people and the environment [6]. Chou et al. invited 12 experts who practice Qigong, Tai Chi, and meditation to be representative participants who have strong perceptions regarding environmental Qi flow, the factors that influence a "healthy environment," and the Qi experience [6]. They attempted to interpret the intensity of the quality of the different natural spaces and to determine the factors that affect the health quality of the environment. The qualitative results included 47 questions. To meet the requirements of content validity, we invited three mind–body exercise experts to review the questionnaires. The final TEQ version was extracted from an in-depth semi-structured interview that included 35 questions using a 5-point Likert scale, which presented the concept of Eastern environmental perception in human–environment interactions to measure the quality of the environment. The questionnaire shows acceptable levels of internal reliability ($\alpha = 0.96$). The questions dealt with the following factors: (1) landscape structure, (2) vegetation characteristics, (3) brightness, (4) visual quality, (5) microclimate, (6) disturbances, and (7) healthy feelings. The higher the score given by participants, the better the quality of TEQ they felt in the environment.

### 2.2.2. Qi Experience(QE) Questionnaires

Qi experience questionnaires were created by Hung, Hwang, and Chang [30] to describe the inner feelings of Qigong practitioners. Responses to these questionnaires were encoded as follows: (1) feeling of Qi, (2) mind, (3) Qi and consciousness, (4) physical, mental, and spiritual benefits, and (5) feeling of Tao [30,31]. Tao is a concept involving human beings following the laws of nature within the traditional Eastern concept of there being a relationship between people and the environment. Statements relating to feelings of Qi, for instance, included: "In this practice, I feel a good sense of Qi". Statements relating to mind included: "In this practice, I feel tranquil" and "After practicing, I feel that my stress has been released". Statements relating to Qi and consciousness included: "In this practice, I can focus on my consciousness". Statements relating to physical, mental, and spiritual benefits included: "After practicing, my body and mind feel completely refreshed". Statements relating to the feeling of Tao included: "In this practice, I can feel the meaning of 'Tao follows nature'" and "In this practice, my spirit is clear and has a deep understanding of nature". The internal reliability score (Cronbach's Alpha) is 0.972. The questionnaire used a 5-point Likert scale to measure participants' Qi experience in practice. The higher the score participants gave, the more Qi experience they perceived.

### 2.2.3. Flow State Scale (FSS)

Flow, as a concept, was proposed by Csikszentmihalyi [32]. Flow describes the feeling of balancing both a high level of skill and challenge. A person in a state of flow feels action and awareness merging, or a sense of losing consciousness, and may even feel a sense of merging with the environment, a state that is also called "optimal experience". The Flow State Scale (FSS) was developed by Jackson and Marsh [33] to measure flow experience. In this study, we translated the English version into Chinese and modified its sentences to describe flow experience in relation to Qigong [30]. The internal reliability score (Cronbach's Alpha) is 0.976. The higher the score participants gave, the deeper the flow experience they perceived.

### 2.2.4. Restorative Experience (RE)

In attention restoration theory (ART), natural elements, such as animals, plants, or water, reflect our state of mind and stimulate our visual senses [16,34]. Complementing this, Davis [35] notes that when one reaches the state of optimal experience, he or she may go into a state of self-reflection. Therefore, we used two questions based on a 5-point Likert scale proposed by Herzog, Black, Fountaine, and Knott [36] to understand which types of environment on a green urban campus may affect Qigong practitioners' restorative experience. The higher the score, the better the participants' experience of recovery and reflection in the given environments.

### 2.2.5. Preference (PRE)

Preference is an overall evaluation of landscape that relates to the information, personal perceptions, and emotional attachment found in the environment. The degree of preference is a reaction of overall satisfaction during connection with the environment. We used one question on a 5-point Likert scale to measure the types of environment found on a green urban campus in order to understand degrees of preference. The higher the preference ranking, the greater the preference for each type of environment.

### 2.3. Research Steps and Participants

The research steps were as follows: Step 1—We invited participants from the primary Qigong education course at NTU and the Taipei Tan Tao Culture Research Association (a civil society official organization involved in the research and teaching of Qigong to citizens). Step 2—Before joining the project, the researcher ensured that participants were over 20 years old and confirmed their willingness and benefits of participation, which were in line with the research ethics statement. The participants were asked to sign the informed

consent form. Step 3—Each participant ($N$ = 58, $M$ = 33.7 years old) was randomly assigned to practice Qigong at 12 experimental sites at NTU between 7 and 11 a.m. This is the time during which Kendall [1] considers that direct contact between the human body and internal organs and sunlight may produce good Qi flow in the body. Step 4—After practicing breathing for 10 min, finishing the questionnaires, and returning the Qi-questionnaires package to the researchers, participants were given a gift as a reward. The questionnaire order started with psychological activity states (flow and Qi experience), recovery and reflection experience about the environment, the perceived traditional environmental Qi, and landscape preference, followed by personal information (i.e., gender, age, years of practicing Qigong).

## 3. Results

### 3.1. A Description of the Data

A total of 654 valid data points were collected. They showed no significant difference between males and females in a flow experience, TEQ, recovery experience, or preference. Slight differences were found in Qi experience [$t$ = 2.081, $df$ = 652, $p$ = 0.038, $\eta^2$ = 0.007] and reflection experience [$t$ = 2.903, $df$ = 462.055, $p$ = 0.004, $\eta^2$ = 0.012]. The results of the ANOVA showed no group difference of years of practicing Qigong in flow experience, recovery experience, reflection experience, or preference. However, Qi experience [$F(2, 651)$ = 5.044, $p$ = 0.007, $\eta^2$ = 0.015] is significantly positively correlated with years of practicing Qigong. From the post hoc for Qi experience, those who practiced for more than 6 years ($M$ = 3.737, $SD$ = 0.805) had a significantly better Qi experience than those who practiced for fewer than 5 years ($M$ = 3.465, $SD$ = 0.724).

### 3.2. Significant Differences in Experiences between Landscape Types in a Green Urban Campus

The $F$ tests show that the landscape types had a significant influence on the experiences in the green urban environments, as follows: flow experience $F(6,647)$ = 5.992, $p$ = 0.000, $\eta^2$ = 0.053; Qi experience $F(6,647)$ = 9.007 $p$ = 0.000, $\eta^2$ = 0.077; recovery experience $F(6,647)$ = 16.083, $p$ = 0.000, $\eta^2$ = 0.130; reflection $F(6,647)$ = 14.868, $p$ = 0.000, $\eta^2$ = 0.121; TEQ $F(6,646)$ = 21.716, $p$ = 0.000, $\eta^2$ = 0.168; and preference $F(6,647)$ = 17.270, $p$ = 0.000, $\eta^2$ = 0.138. According to the report of eta-squared effect size values, this study shows that landscape types have a significant effect on TEQ. Table 1 and Figures 2–4 show the details of this significant influence on experiences in a green urban environment. Table 1 shows each sample of our experimental sites, variables, and the rating by the mean score in the green urban spaces. The results show that flow experience and Qi experience among tree-covered landscapes, waterscapes, and meadows were significantly different from experiences in plazas. Recovery and reflection experience, TEQ, and preference for waterscapes, meadows, tree-covered landscapes, and plazas with waterscapes were significantly better than for plaza landscapes.

**Table 1.** The significant differences between experiences in various landscape types in green urban spaces.

| | W($n$ = 87) | | | M ($n$ = 129) | | | T ($n$ = 89) | | | P ($n$ = 87) | | | PW ($n$ = 65) | | | PM ($n$ = 87) | | | PT ($n$ = 110) | | | F | Post Hoc |
|---|---|---|---|---|---|---|---|---|---|---|---|---|---|---|---|---|---|---|---|---|---|---|---|
| | *M* | *SD* | R | *M* | *SD* | R | *M* | *SD* | R | *M* | *SD* | R | *M* | *SD* | R | *M* | *SD* | R | *M* | *SD* | R | | |
| FE | 3.572 | 0.574 | 3 | 3.597 | 0.530 | 2 | 3.646 | 0.582 | 1 | 3.214 | 0.526 | 7 | 3.506 | 0.598 | 4 | 3.455 | 0.585 | 5 | 3.396 | 0.643 | 6 | 5.99 *** | W > P * M > P ** T > P *** |
| QE | 3.649 | 0.691 | 3 | 3.726 | 0.626 | 2 | 3.764 | 0.716 | 1 | 3.171 | 0.622 | 7 | 3.589 | 0.718 | 4 | 3.446 | 0.660 | 5 | 3.381 | 0.776 | 6 | 9.01 *** | W > P ** M > P *** M > PT * T > P *** T > PT * PW > P * |
| REC | 3.724 | 0.773 | 2 | 3.636 | 0.847 | 3 | 3.820 | 0.833 | 1 | 2.816 | 0.843 | 7 | 3.462 | 0.849 | 4 | 3.322 | 0.883 | 5 | 3.118 | 0.946 | 6 | 16.08 *** | W > P *** W > PT ** M > P *** M > PT ** T > P *** T > PM * T > PT *** PW > P ** PM > P * |
| REF | 3.713 | 0.848 | 1 | 3.543 | 0.829 | 3 | 3.708 | 0.979 | 2 | 2.759 | 0.806 | 7 | 3.492 | 0.812 | 4 | 3.218 | 0.908 | 5 | 3.073 | 0.955 | 6 | 14.87 *** | W > P *** W > PM * W > PT *** M > P *** M > PT * T > P *** T > PM* T > PT *** PW > P *** |
| TEQ | 3.785 | 0.523 | 2 | 3.737 | 0.517 | 3 | 3.804 | 0.590 | 1 | 3.164 | 0.496 | 7 | 3.682 | 0.520 | 4 | 3.360 | 0.592 | 5 | 3.290 | 0.601 | 6 | 21.72 *** | W > P *** W > PM *** W > PT *** M > P *** M > PM ** M > PT *** T > P *** T > PM *** T > PT *** PW > P *** PW > PT ** PW > PM * |
| PRE | 3.862 | 0.750 | 2 | 3.798 | 0.823 | 3 | 3.921 | 0.882 | 1 | 2.954 | 0.875 | 7 | 3.754 | 0.771 | 4 | 3.264 | 0.933 | 6 | 3.273 | 0.976 | 5 | 17.27 *** | W > P *** W > PM *** W > PT *** M > P *** M > PM ** M > PT *** T > P *** T > PM *** T > PT *** PW > P *** PW > PM * PW > PT ** |

N = 58; $n$ = 654, $p < 0.001$ ***, $p < 0.01$ **, $p < 0.05$ *. Note: 1. The rating (R) by the mean of the variables; 2. Variables: FE = Flow experience; QE = Qi experience; REC = Recovery experience; REF = Reflection experience; TEQ = Traditional environmental Qi; PRE = Preference; Waterscapes (W); Meadow (M); Tree-covered (T); Plazas (P); Plazas with Waterscapes (PW); Plazas with Meadow (PM); Plazas with Trees (PT); 3. In the statistical analysis of Levene's test, all variables passed the test for homogeneity except "Preference." Therefore, we used the Brown–Forsythe test to test the homogeneity of preference and used the Games-Howell test to do the post hoc analysis.

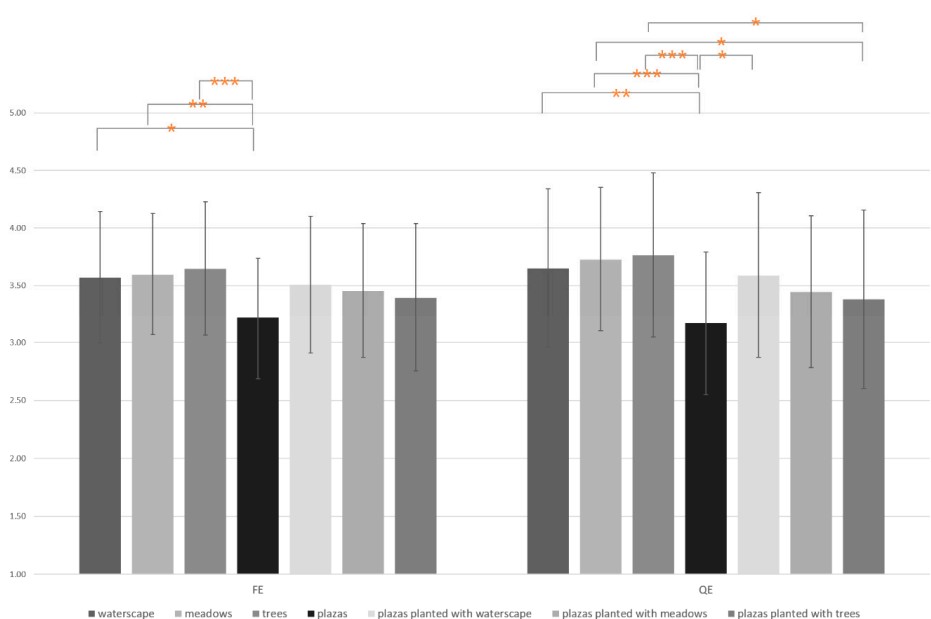

**Figure 2.** Flow experience and Qi experience were significantly different between landscape types ($p < 0.001$ ***, $p < 0.01$ **, $p < 0.05$ *). This figure describes the pattern in experiences among tree-covered landscapes, waterscapes, meadows, plazas planted with meadows, plazas with waterscapes, plazas planted with trees, and plazas. (**a**) There was no significant difference among waterscapes, meadows, and tree-covered landscapes, but when we added plazas with waterscapes, meadows, and trees, the average score of flow experience and Qi experience went up. There was a significant difference between landscape types in regard to flow and Qi experience.

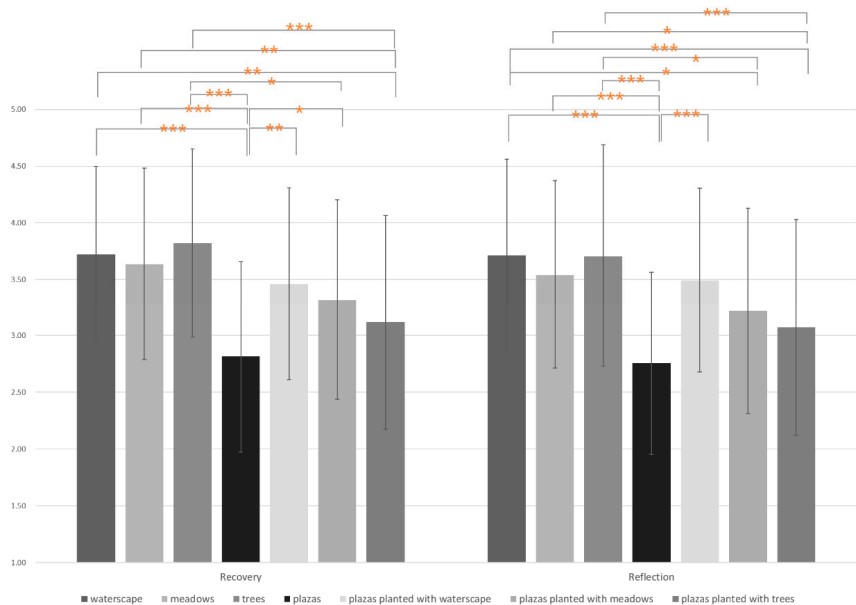

**Figure 3.** Recovery and reflection experiences were significantly different between landscape types ($p < 0.001$ ***, $p < 0.01$ **, $p < 0.05$ *). This figure describes the pattern in experiences among tree-covered landscapes, waterscapes, meadows, plazas planted with meadows, plazas with waterscapes, plazas planted with trees, and plazas. (**b**) There was no significant difference among waterscapes, meadows, and trees, but when we added plazas with waterscapes, meadows, and trees, the average score of recovery and reflection went up. There was a significant difference between landscape types in regard to recovery and reflection.

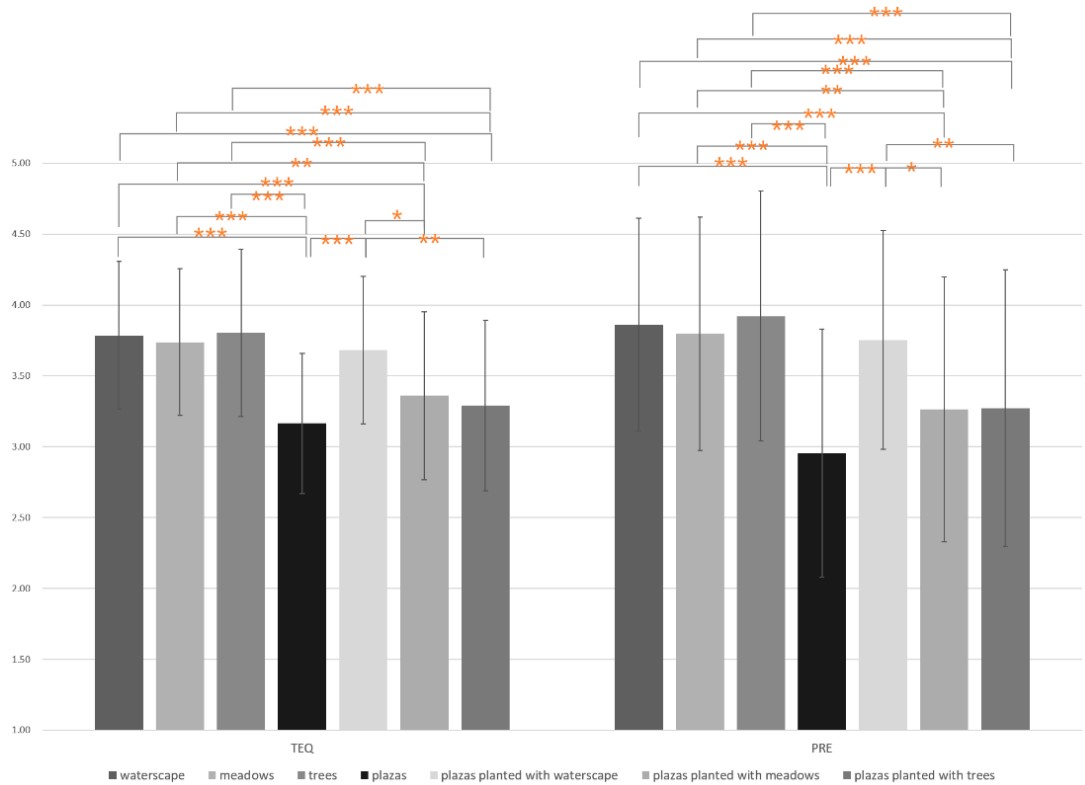

**Figure 4.** Traditional environmental Qi (TEQ) and preference were significantly different between landscape types (*p* < 0.001 ***, *p* < 0.01 **, *p* < 0.05 *). This figure describes the pattern in experiences among tree-covered landscapes, waterscapes, meadows, plazas planted with meadows, plazas with waterscapes, plazas planted with trees, and plazas. (**c**) There was no significant difference among waterscapes, meadows, and trees, but when we added plazas with waterscapes, meadows, and trees, the average score of TEQ and preference went up. There was a significant difference between landscape types in regard to TEQ and preference.

From the post hoc for landscape types, first, waterscapes scored significantly better in flow experience (FE) (*M* = 3.572, *SD* = 0.574), Qi experience (QE) (*M* = 3.649, *SD* = 0.691), recovery (REC) (*M* = 3.724, *SD* = 0.773), reflection (REF) (*M* = 3.713, *SD* = 0.848), traditional environmental Qi (TEQ) (*M* = 3.785, *SD* = 0.523), and preference (PRE) (*M* = 3.862, *SD* = 0.750) than plazas, plazas with meadows, and plazas with trees did (Table 1). Moreover, Qigong practitioners who stood at the viewing platform beside the waterscape experienced a high flow experience (FE) (*M* = 3.806, *SD* = 0.589), Qi experience (QE) (*M* = 3.901, *SD* = 0.703), recovery (REC) (*M* = 4.000, *SD* = 0.632), reflection (REF) (*M* = 3.810, *SD* = 0.750), traditional environmental Qi (TEQ) (*M* = 4.020, *SD* = 0.520), and preference (PRE) (*M* = 4.190, *SD* = 0.602) (Figure 1). We considered waterscapes to be a healthy environment for Qigong practitioners to engage in recovery and self-reflection during interaction with traditional environmental Qi. Second, the meadow landscapes were selected because they provide a sense of openness or enclosure for Qi practitioners. Experiences at these sites were described as follows: FE (*M* = 3.597, *SD* = 0.530), QE (*M* = 3.726, *SD* = 0.626), REC (*M* = 3.364, *SD* = 0.847), REF (*M* = 3.543, *SD* = 0.829), TEQ (*M* = 3.737, *SD* = 0.517), and PRE (*M* = 3.798, *SD* = 0.823) (Table 1). The best meadow places, overall, were at the experimental farm, which is an urban farm landscape with open views that induce the experience of recovery (*M* = 3.800, *SD* = 0.768) and reflection (*M* = 3.800, *SD* = 0.834) (Figure 1). Third, the ratings for tree-covered landscapes were the best, for instance, FE (*M* = 3.646, *SD* = 0.582), QE (*M* = 3.764, *SD* = 0.716), REC (*M* = 3.820, *SD* = 0.833), REF (*M* = 3.708, *SD* = 0.979), TEQ (*M* = 3.804, SD = 0.590), and PRE (*M* = 3.921, *SD* = 0.882) (Table 1). In addition, standing in a tree-covered area near a waterscape but without a view of the water provided a semi-open space that supported the best experiences in a flow experience,

Qi experience, recovery experience, traditional environmental Qi, and preference (Figure 1). Tree-covered landscapes seemed to significantly affect those experiences.

Plazas were selected to present a contrast between urban green spaces and urban buildings. All the mean experience scores were lower than 3.22 (Table 1). Practicing Qigong in front of the plaza was the worst environment chosen among these experimental sites, showing scores of FE ($M = 2.839$, $SD = 0.475$), QE ($M = 2.692$, $SD = 0.511$), REC ($M = 2.238$, $SD = 0.700$), REF ($M = 2.143$, $SD = 0.727$), TEQ ($M = 2.863$, $SD = 0.368$), and PRE ($M = 2.190$, $SD = 0.680$) (Figure 1). The arrangement of limited plants, hard pavement, and roads connecting to the plaza environments may have disturbed the Qigong practice. However, combining a plaza with natural elements, for example, water, meadows, and trees, raised the mean score for all experiences (Table 1). The overall plaza with waterscapes produced good traditional environmental Qi ($M = 3.682$, $SD = 0.520$) for Qigong participants. In addition, preference ($M = 3.754$, $SD = 0.771$) for a plaza with waterscapes was high (Table 1). The health benefits of connection with natural, biophilic elements for human beings have been widely discussed in the literature, and thus, our results are not only in line with previous research, but also integrate with the concept of TEQ and Qi experience to engage with the Eastern environmental perception of landscape arrangements that influence our psychological outcomes.

## 4. Discussion

We have shown that urban green spaces play a critical role in supporting flow and Qi experiences, especially in waterscapes, meadows, and tree-covered landscapes. Our study aims to discuss which landscape types in urban green spaces influence those experiences. Based on environmental psychological theories related to human–environment relationships, our findings are in line with attention restoration theory, preferences, stress reduction theory, and biophilia hypothesis, which all argue that a restorative environment and natural, rather than urban, environmental elements can help people recover from mental fatigue and induce positive emotions.

### 4.1. Urban Landscape Types Influence Experience

Our study found that the impact of landscape types in a green urban environment on experience (FE., QE., TEQ., REC., REF., & PRE) is significant. Chou et al. [30] emphasize that good TEQ, as perceived by human beings, includes brightness, less interference, good configuration, and composition that includes natural elements and plants. Kellert [37] says that places with biophilic design foster feelings of engagement, immersion, emotional attachment, and an "authentic" experience of nature; this seems to relate to the sites that we studied with waterscapes, meadows, and tree-covered landscapes, which showed good experiences of Qi and health benefits.

#### 4.1.1. Waterscapes' Influence on Experience

Waterscapes evoke TEQ, recovery, and reflection experiences. Research shows that people prefer water and that it influences the restoration of our attention [15,26]. The waterscapes we selected in the green urban campus are flat with small fountains, brightness, fresh air, and an ecological landscape with appropriate plants and fewer surrounding buildings. These led the Qigong practitioners to feel a direct experience of nature. Mak [38] stated that the elements of water, mountains, valleys, trees, and meadows in landscapes can produce good environmental Qi. Waterscapes may lead Qigong practitioners to think about personal issues and to calm down, allowing them to gain a state of recovery and reflection. Consistent with previous studies, a natural environment not only induces attention restoration but also produces a sense of relaxation and meditation space [14,36,39].

#### 4.1.2. Tree-Covered Landscapes' Influence on Experience

Tree-covered landscapes evoke an experience of flow, Qi, recovery, reflection, and TEQ. Compared to the hard pavement and minimal trees in plazas, tree-covered landscapes

seem to be more natural, comfortable, and suitable for practicing Qigong. Besides, there are more attractive and more biophilic elements in a tree-covered landscape, such as brightness, openness or partial openness, plants, good views, and a landscape that fosters engagement and immersion, leading to a sense of flow, Qi, and a recovery state of mind. In the tree-covered landscapes, the environment provides a sense of prospect and refuge, allowing the practitioner to concentrate on their exercise. Additionally, participants like to sit near the trees, adjust their breathing, and even go into the state of meditation [2,3]. This concurs with the prospect–refuge theory [40], which states that an environment that gives people the capacity to observe (prospect) without being seen (refuge) meets one's personal needs and security. Therefore, a tree-covered landscape promotes better shelter and a sense of safety for Qi practitioners than a plaza landscape.

### 4.1.3. Meadow Landscapes' Influence on Experience

A meadow landscape supports Qi practitioners in attaining an experience of flow and Qi. The urban farm, which is categorized among the meadow landscapes, was the most preferred landscape. It provided a flat setting that is open and bright, thereby inducing a high flow and Qi experience. These findings are in line with related landscape research theories. For instance, the savanna hypothesis states that people prefer a savanna-like terrain that includes scattered trees and copses, which arranges openness, space, and shade that evoke one's affective affiliation [29]. Large areas of grassland with visible views prompt positive psychological and physiological responses in urban open spaces [7]. Chen [2] and Liu [3] both describe meadow or tree landscapes as places where Qigong practitioners like to practice. William and Harvey [11] note that as people become more relaxed from being in an attractive environment, they can feel a deeper sense of flow experience. Moreover, people prefer savanna-type trees that spread across meadow landscapes that offer a good view [29,41].

### 4.1.4. Other Landscapes' Influence on Experience

As may be expected from the above results, our study confirms that the Qigong practitioners have less preference for plaza landscapes, which offer surroundings with limited natural features and with ceramic tiles on the ground. When water, meadow, or tree features were added to a plaza, the ratings for all experiences went up slightly but were still lower than for natural landscapes. Our findings confirm that natural elements in urban green spaces promote a better experience for Qi practitioners. Moreover, exposure to visually coherent, unthreatening, and restorative natural features, including vegetation and water, triggered positive responses [14,17,20–22,26,41]. These positive responses are rooted in biological responses and are deeply connected with nature [19].

### *4.2. Suggestions for Future Research*

Eastern researchers like to use overall health effects to discuss sensations in Tao, Tai Chi, and Qi. In this study, we used signal indicators to predict the overall Qi experience and discuss the relationship between arrangements of traditional environmental Qi in urban green spaces. First, we may have missed factors that influenced our results. However, the results provide helpful information for urban residents to help them choose healthy green spaces for exercising at NTU. Second, readers may question whether the NTU campus is a special site and whether these results are generally applicable. However, the NTU campus is highly representative of real urban green infrastructure sites in Taipei City. Third, much research has been done on heart rate variability in various landscape types, but research on Qi experience and surrounding environments are still rare. Finally, this study focused on the relationship between flow experience, Qi experience, TEQ, psychological outcomes, and landscape types. We are gradually gaining a better understanding of the content of TEQ and explaining its meaning in terms of human health benefits. Moreover, through extending scientific understanding of the Eastern environmental concept of Feng Shui through TEQ, we could build a substantial index of the overall experience for

designers to help them plan healthy TEQ for mind–body exercise in urban green spaces. This seems to be a very important issue for related environmental researchers, planners, and designers to understand the influence of traditional environmental Qi, Qi experience, and flow experience on the health benefits of environments, especially in urban green spaces.

## 5. Conclusions

The purpose of this study was to understand which kind of urban green spaces support the practice of Qigong. The results demonstrate that meadow and tree-covered landscapes in a green urban environment provide space for mind–body exercises, such as Qigong. In our experimental sites, the features of openness, brightness, and safety, which were found especially in meadows and tree-covered landscapes, evoked Qi experience and flow experience. Moreover, waterscapes and tree-covered landscapes ranked best in preference, recovery and reflection, and TEQ. Waterscapes and trees provide the opportunity to recover attention and provoke reflection. Waterscapes produce fresh air and a sense of tranquility for human beings. Natural features, such as vegetation or water in an unthreatening natural landscape, produce positive affective responses. Overall, the scientific results show that waterscapes, meadows, and tree-covered landscapes were more restorative, provided better TEQ, and were better environments for evoking flow and Qi experiences. Moreover, our findings from this evidence-based scientific analysis could be applied to empirical green infrastructure and biophilic design to improve people's experience of Qi and flow and to provide other psychological benefits or perceived health benefits related to environmental attributes in the everyday urban environment.

**Author Contributions:** The following statements of authors contributions: conceptualization, S.-H.H., W.-Y.C., and C.-Y.C.; methodology, S.-H.H., and W.-Y.C.; formal analysis, S.-H.H.; investigation S.-H.H.; writing—original draft preparation, S.-H.H.; supervision, W.-Y.C., and C.-Y.C.; funding acquisition, C.-Y.C. All authors have read and agreed to the published version of the manuscript.

**Funding:** The study was supported in part by the National Taiwan University Cutting-Edge Steering Research Project (Grant Number 10R70613).

**Institutional Review Board Statement:** Ethical review and approval were waived for this study, the research funding from the Ministry of Education was not expected to apply for the research ethics committee (REC) during that time.

**Informed Consent Statement:** Although the ethical review and approval were waived for this study, informed consent was still obtained from all subjects involved in the study.

**Data Availability Statement:** The data presented in this study are available on request from the corresponding author. The data are not publicly available due to privacy concern.

**Acknowledgments:** This journal article is modified parts from the first author master's thesis of Effect of Landscape Types on Flow experience and Qi Experience, Department of Horticulture and Landscape, National Taiwan University. The authors thank subjects who joined our study.

**Conflicts of Interest:** The authors declare no conflict of interest. The funders had no role in the design of the study; in the collection, analyses, or interpretation of data; in the writing of the manuscript, or in the decision to publish the results.

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
