# Peer review of "A Study on Practicing Qigong and Getting Better Health Benefits in Biophilic Urban Green Spaces"

_sustainability, doi:10.3390/su13041692_

Round 1

Reviewer 1 Report

In the Introduction section I miss

  • a few citied articles on the topic of aesthetic values of waters, meadows and forests;
  • a clearly defined research question.

in the Materials and Methods section, a sub-section Area description should be added. A map of the area with clearly shown locations of the types of areas that the authors write about in the article must also be attached.

Reviewer 2 Report

The paper addresses an interesting topic, still there are several issues that need to be corrected and/or clarified on the paper, namely:

Literature review misses several seminal works and important advances both on biophilic design and Green Infrastructure Planning and design. Read for example (Loures et al., or Ferreira et al., regarding this subject).

Material and methods present some flaws, considering not only that it is hard to understand the used methodology, but also because the research steps are not adequately described. Moreover, the envisioned method consideres the use of questionnaires, though not specifying how does randomness is granted, why was this universe defined, and which were the underlaying principles for c ase study selection. These aspects though small are crucial on such a methodological approach...

Additionally some of the results are also not adequately presented. Table 1 and figure 2, for example, present little information regarding the objectives of the research, and are fuzzy and hard to read. Further information is needed and the relation between biophilic design and perceived health benefits is needed, specially considering the nature of the assumed results and conclusions.

The presented aspects need to be supported on further data.

Conclusions need to be more scientific. As they are they highlight the limitations of the research and they appear to be very too empirical for a research paper.

Round 2

Reviewer 1 Report

-

Author Response

Thanks for your time and energy in reviewing our manuscript. We appreciate it. 

Reviewer 2 Report

The introduced changes contributed to highlight the quality of the paper.

Still, a grammar review should be considered, since there are several phrases that are too long and a few typos along the text. 

Author Response

Please see the attachment. Thank you for your time and energy in reviewing our manuscript.
